# Global expansion of tropical cyclone precipitation footprint

Lianjie Qin [1,10], Laiyin Zhu [2,10] ✉, Baoyin Liu [3], Zixuan Li [4], Yugang Tian[5], Gordon Mitchell [6], Shifei Shen[1], Wei Xu [7,8,9] ✉ & Jianguo Chen [1] ✉

Precipitation from tropical cyclones (TCs) can cause massive damage from inland floods and is becoming more intense under a warming climate. However, knowledge gaps still exist in changes of spatial patterns in heavy TC precipitation. Here we define a metric, DIST30, as the mean radial distance from centers of clustered heavy rainfall cells (> 30 mm/3 h) to TC center, representing the footprint of heavy TC precipitation. There is significant global increase in DIST30 at a rate of 0.34 km/year. Increases of DIST30 cover 59.87% of total TC impact areas, with growth especially strong in the Western North Pacific, Northern Atlantic, and Southern Pacific. The XGBoost machine learning model showed that monthly DIST30 variability is majorly controlled by TC maximum wind speed, location, sea surface temperature, vertical wind shear, and total water column vapor. TC poleward migration in the Northern Hemisphere contributes substantially to the DIST30 upward trend globally.

Tropical cyclones (TCs) have a large area footprint and strong destructive power which threatens lives and property[1–3]. TC impact is mainly due to strong winds, storm surges in coastal areas, and heavy precipitation[4,5]. Future climate model projections generally indicate a decrease in the global average frequency of TCs and an increase in the global average intensity of TCs[6]. An increased TC rainfall rate is expected with a warming climate due to the Clapeyron–Clausius scaling of water vapor in the atmosphere[7–11]. However, recent studies based on satellite rainfall measurements[12,13] demonstrate that the TC rainfall rate shows a decreasing trend in the inner-core of the rainband with increases in the outer bands. This reverse pattern could be associated with both atmospheric stability and the higher availability of water vapor triggered by warmer sea surface temperature (SST)[12,13]. TC rainfall area is another key factor to affects TC rainfall distribution, which increases with increasing relative SST[14], defined as the difference between the local SST and the average SST over the tropical oceans. Although many prior studies[12,13,15–17] discuss the changes in TC rainfall

rates and accumulated TC precipitation, we still need more knowledge of the spatial structure of TCs and their connection with climate change. Rainfall is controlled by different mechanisms in different parts of the TC. While convective rainfall dominates in the inner core of the TC driven by atmospheric updraft, the outer TC bands generally have stratiform precipitation[18]. Recent studies[19,20] also find that TC precipitation in the inner core and outer region is controlled by TC intensity and environmental conditions. In addition, the majority of TC precipitation research uses rainfall metrics (e.g., mean or median) based on arbitrary radii (e.g., 100 km and 500 km) from the TC center[6,10,15]. However, different definitions of TC rainfall rate can be challenging for multi-model comparisons[10]. Furthermore, the within radii averaging approach may not be an accurate description of the TC rainfall risk because of the high spatial variabilities in TC rainfall in both the core and outer band areas[21,22].

While there is consensus that the TC rainfall rate will increase with a warmer climate following the Clausius-Clapeyron equation, the

[1]School of Safety Science, Tsinghua University, Beijing, China. [2]School of Environment, Geography, and Sustainability, Western Michigan University, Kalamazoo, MI, USA. [3]Institutes of Science and Development, Chinese Academy of Sciences, Beijing, China. [4]School of Finance, Nankai University, Tianjin, China. [5]School of Geography and Information Engineering, China University of Geosciences, Wuhan, China. [6]School of Geography and Water@leeds, University of Leeds, Leeds, UK. [7]Key Laboratory of Environmental Change and Natural Disaster of Ministry of Education, Faculty of Geographical Science, Beijing Normal University, Beijing, China. [8]State Key Laboratory of Earth Surface Processes and Resource Ecology, Beijing Normal University, Beijing, China. [9]Academy of Disaster Reduction and Emergency Management, Ministry of Emergency Management and Ministry of Education, Beijing Normal University, Beijing, China. [10]These authors contributed equally: Lianjie Qin, Laiyin Zhu. ✉e-mail: laiyin.zhu@wmich.edu; xuwei@bnu.edu.cn; chenjianguo@tsinghua.edu.cn

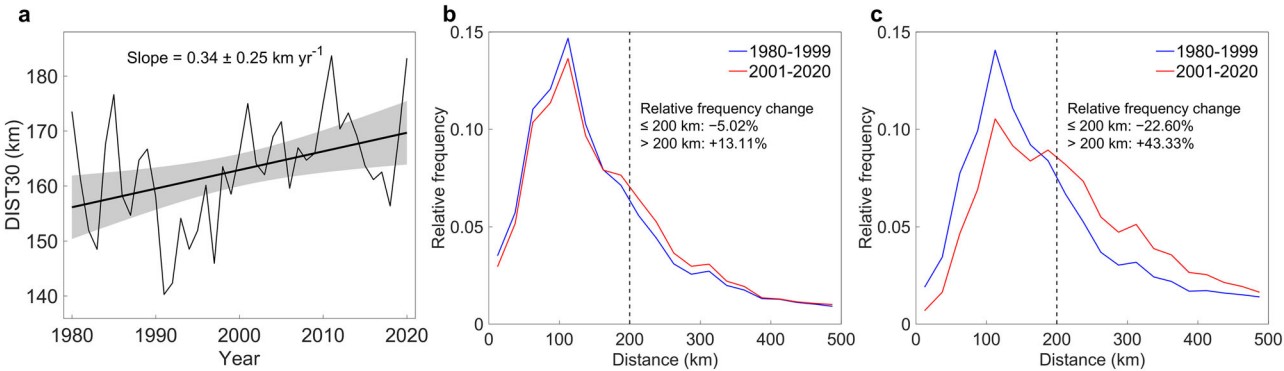

**Fig. 1 | Temporal changes in distance at the threshold of 30 mm/3 hours (DIST30) and tropical cyclone (TC) rainfall. a** Annually time series of globally averaged DIST30. **b** Relative frequency of > 30 mm/3 h precipitation in low latitude (≤ 25°). **c** Relative frequency of > 30 mm/3 h precipitation in mid latitude (> 25°).

Blue lines denote the early period (1980–1999) and red lines denote the late period (2001–2020). Bins with 25 km equal intervals of Distance are created for each 20 years sample. The relative frequency is defined as the frequency of observations in each bin divided by all observations for each 20 years sample.

actual fractional rainfall increase varies above or below the 7% °C⁻¹ theoretical rate with SST warming among different models[23,24]. There is an urgent need to understand mismatches between models and observations, particularly how the fractional rainfall increase will likely change when one moves from the TC center by certain distances[10]. Therefore, here we define a TC rainfall metric DIST30 as the mean radial distance from centers of clustered heavy rainfall cells (> 30 mm/ 3 h) weighted by their rainfall rate (see Methods) to the TC center. As compared with the existing spatial metrics[25–27] for TC rainfall, the DIST30 has a more generalized description of the distance from the center of clustered heavy rainfall cells to each TC center. It provides a more direct approach to understanding the potential major inland flood risk caused by heavy TC rainfall. DIST30 is constructed based on a 41-year quality controlled high-resolution global precipitation data set (MSWEP). Our results indicate that DIST30 has increased at an average rate of 0.34 km/year globally, with stronger increases observed in ocean-land boundary areas and mid-latitudes in the northern hemisphere. Based on our interpretable machine learning algorithm (XGBoost), the TC maximum sustained wind speed (central pressure), location of the TC, SST, vertical wind shear, and total column water vapor are the most important factors. DIST30 shows a particularly strong positive relationship with vertical wind shear in mid-latitudes which explains the recent global increasing trend in DIST30.

## Results

### Global increases in distance from TC center and extreme rainfall

Besides high interannual and interdecadal variabilities, the global annual mean DIST30 (Fig. 1a) demonstrates a statistically significant increasing trend (at a 95% significance level) with a slope of 0.34 km/ year between 1980 and 2020. In addition, we calculated and compared the frequency of extreme TC rainfall (30 mm/3 h) within different bins of distance to each TC center between 1980–1999 and 2000–2020 (Fig. 1b, c). Both low latitudes (≤ 25°) and mid-latitudes (> 25°) observe decreased relative frequency of extreme TC rainfall within 200 km of the TC center. While the low latitude has a 5.02% decrease, the mid-latitude has a larger 22.60% decrease. Conversely, we observe increases in the relative frequency of extreme TC rainfall beyond 200 km from the TC center in both low and mid-latitudes. The low latitudes show a 13.11% increase, but a 43.33% increase in mid-latitudes. To further validate trends found in extreme TC rainfall, we also defined DIST50 as the distance between the TC center and extreme rainfall clusters larger than 50 mm/3 h. DIST50 shows a consistent upward trend of 0.36 km/year (Supplementary Fig. 1a) similar to DIST30, with a smaller change in the relative frequency in low latitudes (−3.02% beyond 200 km from the TC center and +1.12% within 200 km of the

TC center; Supplementary Fig. 1b), but the higher relative change in mid-latitudes (+ 52.52% beyond 200 km from the TC center and − 21.54% within 200 km of the TC center; Supplementary Fig. 1c).

### Spatial distribution of changes in DIST30

To further investigate the spatial variations of DIST30, we calculated temporal differences of DIST30 between the late period (2001–2020) and the early period (1980–1999) (ΔDIST30) for every 4° grid cell (Fig. 2). The results show that DIST30 increases for 59.87% (8.79 × 10⁷ km²) and only decreases for less than 40.13% (5.89 × 10⁷ km²) of the total TC impact areas (Fig. 2 and Supplementary Table 1). The areas with positive ΔDIST30 greater than 25, 50, 75, and 100 km take 39.03% (5.73 × 10⁷ km²), 22.06% (3.24 × 10⁷ km²), 12.01% (1.76 × 10⁷ km²), and 7.41% (1.09 × 10⁷ km²) amount of the global total TC impact areas, respectively (Supplementary Table 2). Among six ocean basins, only the Northern Indian basin experiences a general reduction in DIST30. In the other five ocean basins, increasing trends of DIST30 are more frequent than decreasing trends. The Western North Pacific basin has the largest proportion of areas with DIST30 growth (2.20 × 10⁷ km², 25.06% of global area with DIST30 increase), and the areas with positive ΔDIST30 greater than 25, 50, 75, and 100 km take 39.76% (1.35 × 10⁷ km²), 21.49% (7.30 × 10⁶ km²), 10.62% (3.61 × 10⁶ km²), and 7.49% (2.54 × 10⁶ km²) of the total TC impact areas in Western North Pacific basin, respectively (Supplementary Table 2). The Northern Atlantic basin has the second largest proportion of areas with DIST30 growth (1.92 × 10⁷ km², 21.80% of global area with DIST30 increase), and the areas with positive ΔDIST30 greater than 25, 50, 75, and 100 km take 43.73% (1.29 × 10⁷ km²), 22.64% (6.68 × 10⁶ km²), 13.79% (4.07 × 10⁶ km²), and 7.36% (2.17 × 10⁶ km²) amount of the total TC impact areas in Northern Atlantic basin, respectively (Supplementary Table 2).

In addition, we examined coastal grid cells (defined as grid cells with both land and sea) and found they have slight growth in DIST30. From a total of 4.73 × 10⁷ km² of land and sea boundary areas, slightly more than half (54.02%, representing 2.56 × 10⁷ km²) present an increasing trend in DIST30. More growths in DIST30 are observed in populated coastal areas including eastern coastal areas of China, Japan islands, Korean peninsula, and eastern Australia, since we have observed that 7.33 × 10⁶ km² (61.84%) of Western North Pacific basin boundary areas and 4.93 × 10⁶ km² (63.55%) of South Pacific basin boundary areas showed increases of DIST30. In general, the areas experiencing increases of DIST30 have elevated risk of heavy precipitation far from the TC center, and they take the majority of the TC impact areas globally, indicating a global expansion of TC precipitation footprint and flood risk. In contrast, we observed a downward trend in DIST30 in the North Indian basin coastal areas (1.67 × 10⁶ km², 32.71% of

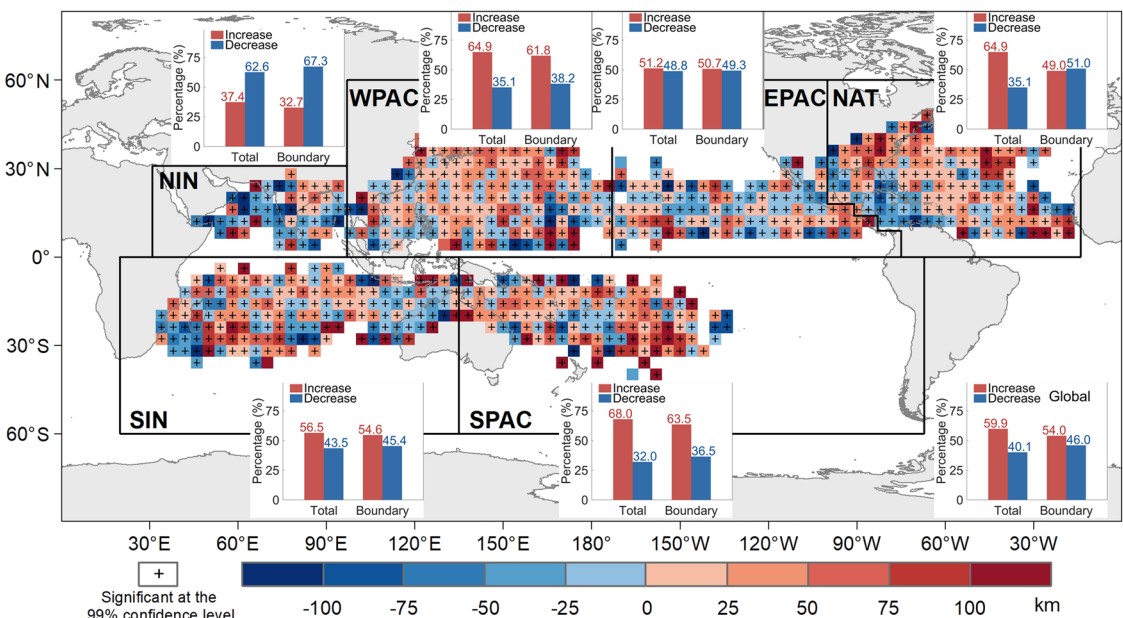

**Fig. 2 | Spatial distribution of differences in distance at the threshold of 30 mm/ 3 hours (DIST30) between the late period (2001-2020) and the early period (1980-1999).** Bars show the percentages of increasing and decreasing areas in each ocean basin/region. The cross indicates the difference at the location passed the Mann-Whitney test on the regression model of time series at the 99% significance level. DIST30 is collected within 4° grid cells for clear visualization of spatial patterns.

area with increasing trends). Other ocean basins only observed slight increases or no trend in DIST30.

## Modeling DIST30 using machine learning

Drawing on high-resolution dynamical General Circulation Models (GCMs) and downscaled Regional Climate Models (RCMs), many studies discuss the sensitivity of TC rainfall rate (Clausius-Clapeyron Scaling) to global climate warming scenarios[9,23,24,28–32]. To understand factors that determine the TC rainfall footprint, we developed XGBoost models for global and ocean basin wide monthly values of DIST30 based on TC characteristics (e.g., central pressure and maximum sustained wind speed) and environmental factors (SST, relative 2-meter air temperature, vertical wind shear). XGBoost is a powerful decision tree-based machine learning approach[33] that repeatedly generates new trees from the initial poor performers ('weak learners') intending to improve the model fit (see Methods for further detail). A variables list is shown in Supplementary Tables 3 and 4. The XGBoost model demonstrates decent out-of-sample prediction performances, with the ratio of the explained variance ($R^2$) = 0.51, root mean square error (RMSE) = 70.48 km, and mean absolute error (MAE) = 47.34 km. Similarly, models for different ocean basins (Supplementary Fig. 2) also show consistent performance ($R^2$ ranging from 0.46 to 0.57, RMSE ranging from 66.66 to 77.90 km, MAE ranging from 43.74 to 50.77 km). The South Indian basin has the highest $R^2$ and the Western North Pacific basin has the lowest RMSE/MAE. All indicate that our XGBoost models have decent prediction performance without overfitting and are reliable sources for model interpretations.

The SHapley Additive exPlanations (SHAP) value[34] (see Methods) is an interpretable artificial intelligence (AI) approach that represents the contribution from each feature to each individual prediction. Therefore, a larger than zero SHAP value indicates a positive contribution from a feature to the mean prediction of the response variable (here DIST30) and vice versa. Here we define the feature importance as the absolute value of all positive and negative contributions of each feature which reflects the sensitivity of DIST30 to the specific feature. Figure 3b demonstrates the 8 top-ranked features for the global DIST30 models. The TC maximum intensity (VMAX) and

Latitude (LAT) rank as the most important variables in the global DIST30 model, followed by the central pressure (PRES), vertical wind shear (WS), and relative 2-meter air temperature (RT2M). In addition, we demonstrated the detailed interactions between DIST30 and four top-ranked features in the XGBoost models (Fig. 3c–f) as well as their spatial distributions (Fig. 4).

DIST30 generally has a negative relationship with VMAX (Fig. 3c), with different sensitivities within different VMAX ranges. Negative contributions to DIST30 (smaller DIST30) are generally associated with a higher VMAX (> 70 knots, ~ Category 1 hurricane intensity). The magnitude of negative contributions of SHAP is stabilized when VMAX is larger than 70 knots. Strong convective precipitation from a well-developed TC system in the tropics is usually located near the eyewalls[35,36], within a short distance to the TC center. In contrast, the DIST30 SHAP increases abruptly when the VMAX is reduced below 70 knots, which could be related to the Extratropical Transition (ET) of TCs. As TCs undergo the ET, warm-core tropical systems change to cold-core mid-latitude cyclones. These mid-latitude systems are generally larger with a more widespread distribution of precipitation than well-developed TCs in the tropics. As the eyewall disappears, the heavy precipitation center for ET storms usually moves further away from the storm center. In Fig. 3d we can observe a clear pattern such that a higher absolute value of LAT is associated with a larger DIST30, whilst mid-latitude regions (20 °N to 40 °N and 20 °S to 40 °S) have more frequent observations of lower VMAX. The translation speed of TC also increases as it moves to higher latitudes[37], contributing to the asymmetry of TC precipitation[21,38] and possibly to the increase in DIST30. In addition to the general positive relationships with latitude, a small number of high DIST30 SHAP values are also located in low latitudes (between 20 °S and 20 °N, Fig. 3b) and have low VMAX. These are possibly associated with scattered tropical thunderstorms in less organized tropical storm systems with weak wind intensity (< 40 knots). The eyewalls of those systems are not well defined, and the center of the thunderstorms with heavy precipitation can be far away from the center.

The vertical wind shear (WS) demonstrates a general positive but non-linear relationship with the DIST30 SHAP values (Figs. 3e and 4c),

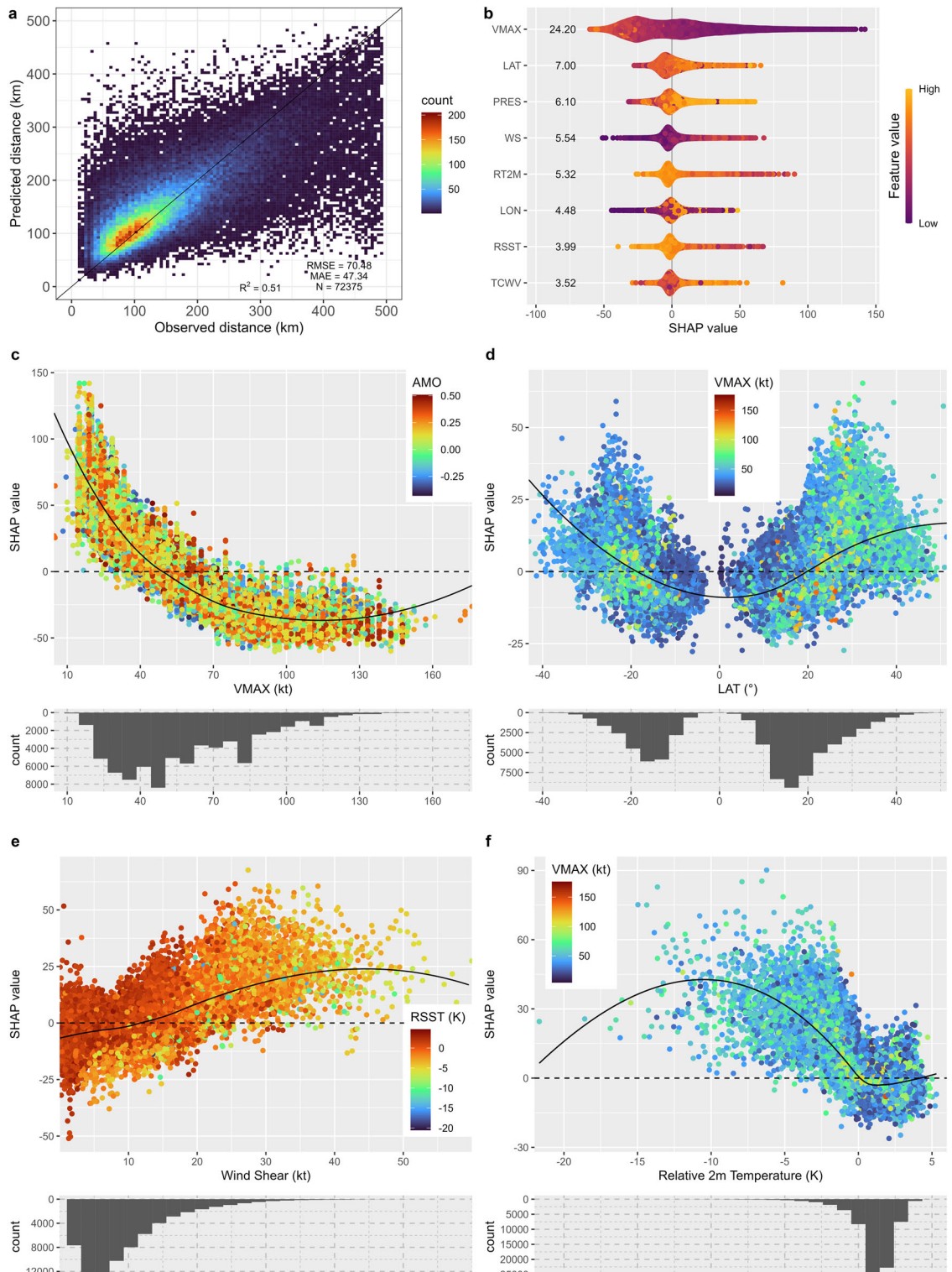

**Fig. 3 | Model performance, feature importance and selected features' relationships with the distance at the threshold of 30 mm/3 hours (DIST30) based on the SHapley Additive exPlanations (SHAP) values from the global XGBoost model.** Models are developed based on monthly averaged DIST30 and environmental variables within 0.25° × 0.25° grid boxes. **a** The scatter density plot of observed and predicted based on 5-fold random cross-validations: one XGBoost model is trained using the data exclusively for each fold and used for prediction based on data from each fold; The predicted values are collected for all five folds and compared with all observed values; color represents the frequency of observations/predictions in each 1/100 bin within the observed DIST30 range. **b** feature importance calculated as the SHAP value for eight top-ranked features with the most importance for the XGBoost model. **c–f** Relationship between the four most important features and DIST30 SHAP value, with distributions of each feature. Dot colors in (**c–f**) denote that the feature has the largest covariance with the main feature (x-axis) in the model.

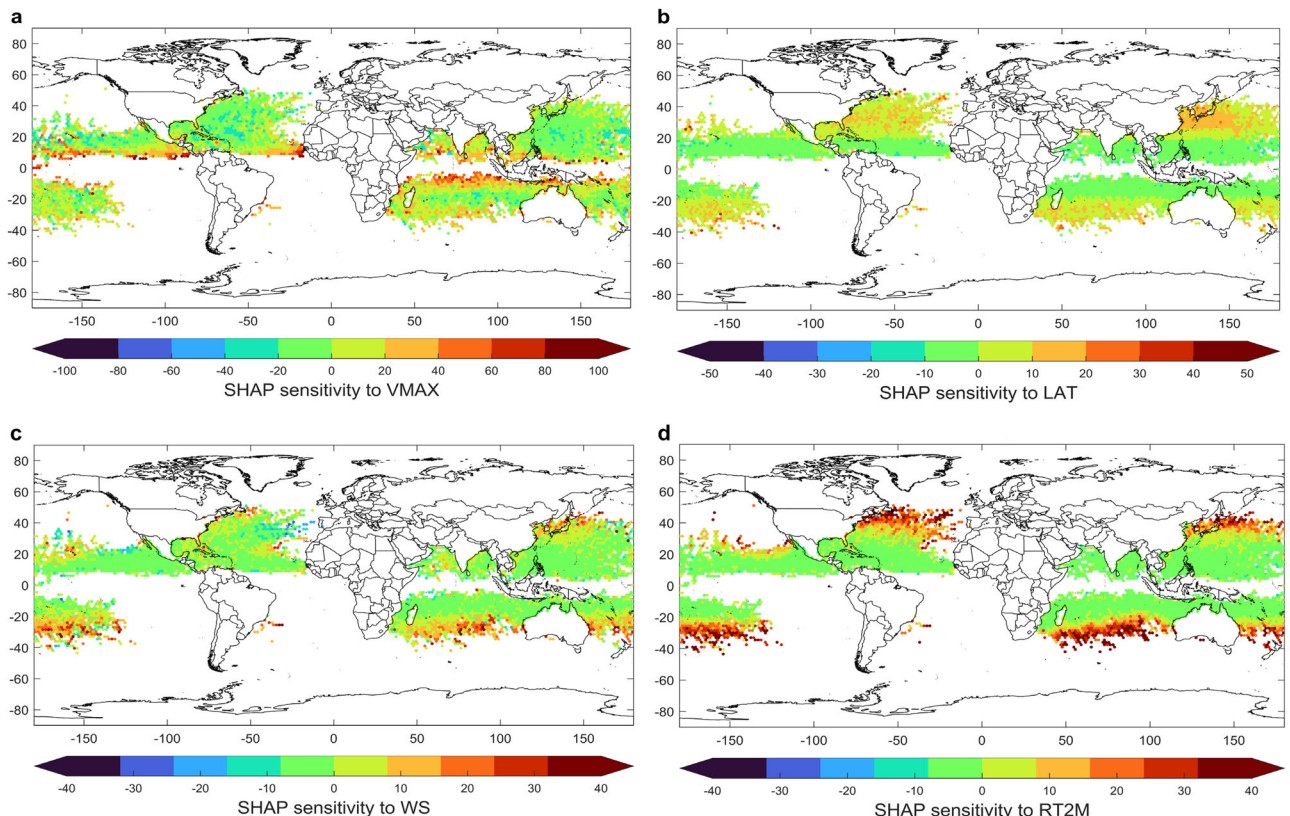

**Fig. 4 | Spatial distribution of SHapley Additive exPlanations (SHAP) values of the four most important variables. a** Maximum wind speed (VMAX). **b** Latitude (LAT). **c** Vertical Wind Shear (WS). **d** Relative Sea Surface Temperature (RSST). Values are averaged to hexagons with radius = 1° for improved visualization.

with complexities when WS is below 15 knots. Higher WS is likely to expand the TC rainfall area and introduce higher rainfall rates far from the TC center[39,40]; this phenomenon is particularly strong in the mid-latitudes when TCs are translating to regions with lower relative sea surface temperature (green and yellow dots in Fig. 3e). More frequent larger WS SHAP values also occur at higher latitudes (Fig. 4c), reflecting the latitudinal gradient. Previous studies have identified WS as one of the key factors that increase the TC precipitation asymmetry[21,38,41,42]. Positive vertical wind shear SHAP values are scattered across different ocean basins in the northern hemisphere (Fig. 4c) and are slightly stronger in the southern hemisphere. The lower range of WS (< 10 knots) demonstrates both negative and positive contributions to the DIST30 SHAP values and those cases are more likely to happen in warmer SSTs (Fig. 3e), indicating a lower sensitivity of DIST30 to the WS in the tropics.

The relative 2-meter air temperature (RT2M) also demonstrates a nonmonotonic relationship with DIST30—a negative relationship when the RT2M is lower than 1 K but a weak positive relationship when the RT2M is higher than 1 K. This non-monotonic relationship could be caused by several factors. The RT2M decreases with latitudes, so larger DIST30 (ET systems) correspond to lower RT2M in higher latitudes, whilst in the tropics, higher RT2M corresponds to higher SST. Higher SST is likely to support more latent heat to the TC system. Interestingly, the SHAP values for the Relative SST (Supplementary Fig. 5h) also demonstrate a positive relationship with DIST30 when it is above 0 K, but the absolute SST has both positive and negative contributions to DIST30 (Supplementary Fig. 5g). Both observations and numerical models have previously demonstrated that the area of TC rainfall increases as the relative SST increases for mature TCs in the tropics, potentially caused by the influences from Relative SST to TC size, relative humidity, and potential intensity[14]. But the TC rainfall area is less sensitive to the absolute SST in the tropics[14]. Our analysis also

indicates that higher total precipitable water vapor (TCWV) is generally associated with larger DIST30 (Supplementary Fig. 5i). This is also consistent with the previous analysis[13], which demonstrates that there is a positive relationship between TCWV and rainfall intensity in the TC outer region, while there is a negative relationship between TCWV and rainfall intensity in the TC inner core. Spatially, a very strong latitudinal gradient is also evident for the SHAP value for RT2M (Fig. 4d).

**Interpreting trends in DIST30**

In previous sections, we have discovered that after TCs' own characteristics (VMAX and PRES), vertical wind shear is the most important environmental variable explaining DIST30, and displays a strong positive but non-linear relationship. Therefore, here we provide additional analysis to explain temporal trends in global DIST30 by splitting our monthly DIST30 data into three parts according to latitude: the tropics between 25 °S and 25 °N, mid-latitudes of the northern hemisphere (> 25 °N), and mid-latitudes of the southern hemisphere (> 25 °S).

In the tropics, DIST30 demonstrates a slight increasing pattern but with low confidence (Fig. 5a, p-value ≈ 0.11). Meanwhile, we identified a statistically significant (p-value ≈ 0.00) decreasing trend in vertical wind shear in the tropics. Lower vertical wind shear is likely to create a more favorable environment for cyclone genesis to survive and evolve into stronger TCs[43,44] in the tropics. Meanwhile, lower vertical wind shear is not likely to directly disperse the rain field and increase the DIST30 in the tropics because of its lower sensitivity in the tropics (SHAP value in Fig. 3e). We did find a weak increasing pattern in the annual frequency of DIST30 in the tropics (p-value ≈ 0.29, Fig. 5c), which could be related to the reduced vertical wind shear. Recent studies have also identified increased SST in the tropics globally[45,46]. Both the more surviving TCs and more latent heat supplied from the ocean surface are likely to contribute to the

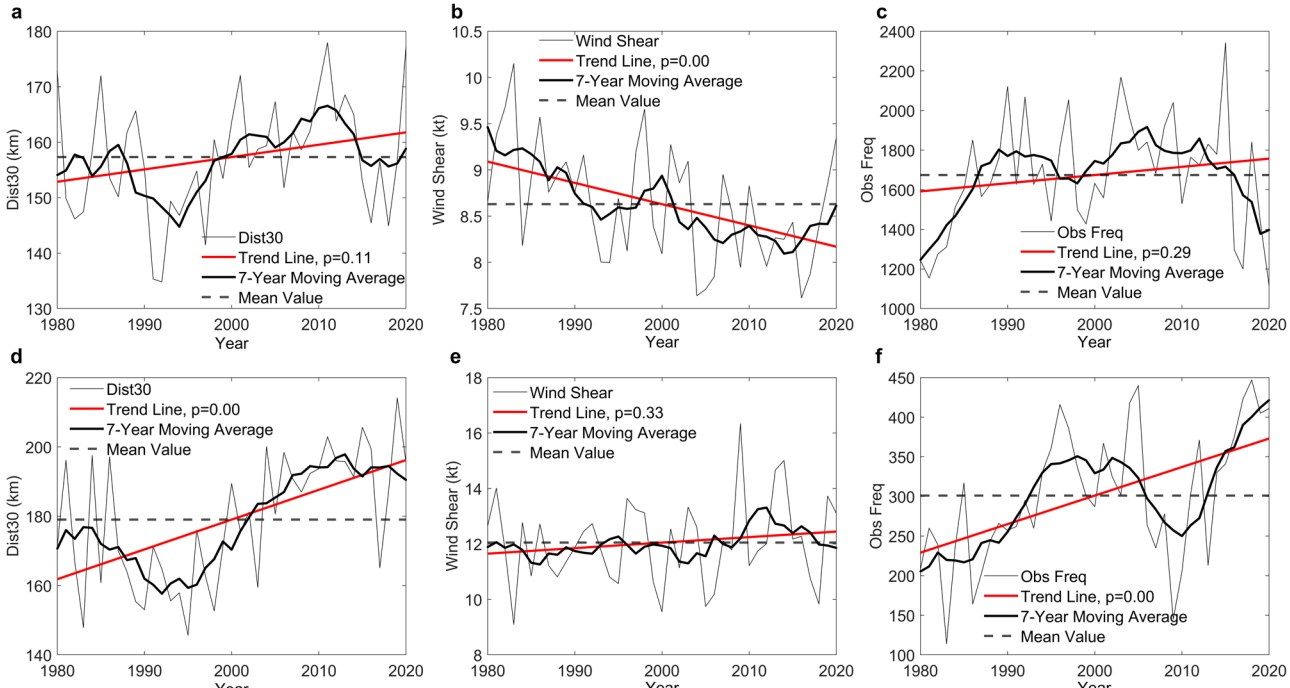

**Fig. 5 | Interpreting the trend in distance at the threshold of 30 mm/3 hours (DIST30). a** Change of DIST30 in the tropics (between 25 °S and 25 °N). **b** Change of vertical wind shear in the tropics. **c** Changes in observation frequency (number of monthly aggregated records) in the tropics. **d** Changes of DIST30 in mid-latitudes of the northern hemisphere (> 25 °N). **e** Changes of vertical wind shear > 25 °N. **f** The changes of observation frequency > 25 °N.

slightly increasing precipitation footprints (DIST30) of TCs within the tropics.

DIST30 in mid-latitudes of the northern hemisphere demonstrates a very strong increasing trend (p-value ≈ 0.00). We believe this is the major contributor to the global increasing trend in DIST30, since both the tropics and southern hemisphere mid-latitudes (Supplementary Fig. 7) demonstrate weak increases of DIST30. The strong increasing trend of DIST30 beyond 25 °N can be explained by changes in both vertical wind shear and the frequency of TCs entering the mid-latitudes. In contrast to the decreasing pattern in the tropics, the vertical wind shear in the northern hemisphere at higher latitudes demonstrates a weakly increasing signal (Fig. 5e, p-value ≈ 0.33). More importantly, a very strong increase is evident for the annual DIST30 observation frequency beyond 25 °N (Fig. 5f, p-value ≈ 0.00). It is probable that more TCs survived beyond 25 °N, translated toward mid-latitudes, and then encountered relatively higher vertical wind shear. Those TCs undergoing extratropical transition are usually associated with more asymmetric distribution of rain fields and enlarged DIST30, which was also demonstrated by the high sensitivity of DIST30 to WS in mid-latitudes from our SHAP analysis (Fig. 3e). Interestingly, we only observe those strong temporal trends in DIST30 in the higher latitudes of the northern hemisphere. In the southern hemisphere, no statistically significant trend has been identified in DIST30, vertical wind shear, and observation frequency beyond 25 °S (Supplementary Fig. 7). The poleward migration of TCs was first described by Kossin et al. [47] and has been discussed by recent studies[48–52]. The poleward migration of TC intensity can be explained by the expansion of the Inter Tropical Convergence Zone (ITCZ) and the tropical Hadley circulation under a warming climate, which changes the environment for TC genesis and development, including the vertical wind shear, SST, and steering flows. Some studies identified a more distinct poleward migration of TCs in the Western North Pacific basin[48,50,53]. Here we discovered that the vertical wind shear in the tropics has become more favorable to TC genesis and evolution globally in the last 41 years. More TCs moved to higher latitudes in the northern hemisphere and interacted with stronger vertical wind shear. Those processes could work together and cause substantial increases of larger TC precipitation footprint (DIST30) globally. Supplementary Figs. 3–6 also show SHAP relationships between all included features and the global DIST30 from our XGBoost modeling.

## Discussion

DIST30 can serve as an important indicator to understand potential flood impacts. A higher value of DIST30 indicates that the most extreme precipitation related to the TC occurs farther away from the center. It is important to note that DIST30 is a numerical value calculated solely from the TC precipitation profile including both the inner core and outer rain bands, as the cases show in Supplementary Fig. 8. We acknowledge that DIST30 can only partially describe the complexity of TC precipitation structure, and future work should focus on optimizing the definition for the TC precipitation structure by including both the rainfall area[19,20,25–27,40] and intensity.

From 1980 to 2020, DIST30 shows an increasing trend of 0.34 km/ year. The relative frequency of > 30 mm/3 h TC precipitation beyond 200 km from the TC center increases by 13.11% at low latitudes and 43.33% at mid-latitudes, while the frequency of > 30 mm/3 h TC precipitation within 200 km of the TC center decreases by 5.02% at low latitudes and 22.60% at mid-latitudes. Spatially, we observe the DIST30 increases in 59.87% ($8.79 \times 10^7$ km²) of the global total TC affected area ($1.47 \times 10^8$ km²). There are $3.24 \times 10^7$ km² with >100 km increases in DIST30, representing 22.07% of the total TC affected area. In addition, 54.02% ($2.56 \times 10^7$ km²) of the global land and sea boundary area ($4.73 \times 10^7$ km²) affected by TCs observed growth in DIST30, while more percentage of areas with DIST30 growth are shown in the Western North Pacific basin (61.84% from a total of $1.19 \times 10^7$ km² affected area) and the Southern Pacific basin (63.55% from a total of $7.76 \times 10^6$ km² affected area).

The XGBoost machine learning model provides skillful out-of-sample predictions for monthly values of DIST30 both globally and ocean basin wide based on environmental variables and climate index.

The models show consistently good performance with high $R^2$ values (0.51 globally, and 0.46–0.57 at different ocean basins) and low RMSE values (70.8 km globally and 66.66–77.90 km at different ocean basins). The interpretable AI approach (SHAP values) identified the most important environmental factors and their relationships with DIST30. Specifically, we find that maximum sustained wind speed (central pressure), latitude, vertical wind shear, relative 2-meter air temperature, relative SST, and total column water vapor are the most important variables for DIST30 in the XGBoost model.

Our models demonstrate that the vertical wind shear is the environmental forcing variable with the most significant influence on DIST30, particularly in the mid-latitudes. Previous studies demonstrated the strong vertical wind shear as one of the most important factors to change the shape and increase the size of the rain field[54,55]. Meanwhile, decreases in vertical wind shear in the tropics likely created a more favorable environment for TC genesis and development, together with the expansion of the ITCZ[47], increasing the number of TCs translating into the mid-latitudes > 25 °N. The increases in TC frequency along with the slightly increased vertical wind shear for regions > 25 °N, largely contributed to the global increase of DIST30. The strongest increasing pattern of DIST30 of all occurs in the Western North Pacific Ocean basin, which agrees with the shifting poleward trends in TCs evident from previous observation-based analysis[53].

Besides the vertical wind shear, the interpretable AI approach identified other possible physical mechanisms controlling DIST30. Both the maximum sustained wind speed and latitude show negative relationships with DIST30. There is negative relationship between DIST30 and the maximum sustained wind speed but this relationship is non-monotonic. The stabilized low DIST30 with high maximum sustained wind speed can be explained by more organized TCs with stronger intensity and strong precipitation in the eye walls near the TC center. However, there is a strong negative relationship between DIST30 and the lower-range maximum sustained wind speed. Possible explanations for this are that unorganized tropical storms and ET TC systems (both with lower wind intensity) usually have rain bands further away from the center. The negative relationship with latitudes is attributed to several factors: 1) TCs translating to higher latitudes move faster therefore their precipitation footprint is larger; 2) the scale of TC undergoing ET is larger than matured TCs in the tropics; 3) TC rain field enlarging effect occurs with more intense vertical wind shear in mid-latitudes. Some variables also show non-linear or non-monotonic relationships with DIST30, including the relative SST and the SST. They both display a strong gradient with latitude, and many larger DIST30 cases correspond to lower air temperature and SST observations in mid-latitudes. In the tropics, we observe that DIST30 slightly increases with relative SST because of its control on size, relative humidity, and potential intensity for mature TCs[14]. There is also a generally positive relationship between DIST30 and the availability of water vapor[12].

There is a high certainty that global warming will increase both air temperature and SST, and therefore make intense TCs more frequent[56,57]. Meanwhile, rising air temperature will increase the atmosphere's capacity to hold more moisture and therefore increase the rain rate from TCs[10]. Our results indicate that footprints of heavy rainfall from TCs have slightly expanded spatially in the tropics but substantially increased in the mid-latitudes of the northern hemisphere, particularly in the Western Pacific and North Atlantic basins. Multiple recent studies[12,13] also demonstrated increases of TC rainfall in the outer rain bands globally. Notably, some recent extreme precipitation events in the mid-latitude generated by TCs were far away from the TC center, including the 2021 Henan flood event from Typhoon In-fa (No. 2106) and the 2023 Hebei flood event from Typhoon Doksuri (No. 2305). Both the scientific community and risk managers need to pay more attention to this spatial migration of TCs and the elevated flood risk related to heavy rainfall from TCs,

particularly for the densely populated communities in the mid and high latitudes of the Northern Hemisphere. Those vulnerable communities are historically less exposed to TC hazards, and so are both physically and psychologically less prepared for TCs. Multiple recent studies[58–62] indicated that their exposure to TC hazard has increased substantially in recent years, posing a big threat to local society. Further work is needed to understand the details of regional mechanisms controlling the changing patterns of TC rainfall in different ocean basins and how their spatial footprints are likely to change in the future warming world.

## Methods

### Data
The International Best Track Archive for Climate Stewardship (IBTrACS) v04 dataset[63] was acquired for the period 1980 to 2020, which includes TC position, minimum sea-level pressure, and the maximum sustained wind speed. The TC-related precipitation data were drawn from the Multi-Source Weighted-Ensemble Precipitation (MSWEP) v2 dataset, which takes advantage of the complementary strengths of in-situ-, satellite-, and reanalysis-based data to provide reliable estimates of precipitation on a global scale[64]. The MSWEP v2 data provides a complete record of precipitation over both ocean and land after 1980 and its length matches with the satellite-based TC location records. The land and sea boundary shape file were obtained from the UCLA Geoportal (see Data availability section).

### TC information and precipitation extraction
All TCs with wind speeds in the best track dataset on both land and ocean during 1980-2020 are used in this study, but only including records at 00:00, 03:00, 06:00, 09:00, 12:00, 15:00, 18:00, and 21:00 UTC. The MSWEP dataset covers the period 1980-2020 with a temporal resolution of 3 h. Here the Precipitation of TCs is defined as accumulated 3-h MSWEP precipitation within 500 km of the TC center[15]. To facilitate the calculation of the distance from the TC center, we first adjust the latitude values (rounding the latitude up for the Northern Hemisphere and down for the Southern Hemisphere) and resample the TC precipitation field to 25 km resolution using the Albers projection. In the Albers projection, standard parallels are set to the adjusted latitude of TC center ± 10°. The number of rows and columns of the TC precipitation field after resampling is 41, and the coordinates of the TC center are (21,21), which we note here as the TC center $(X_0, Y_0)$.

### DIST30 and DIST50
DIST30 is the distance from the center of clustered heavy rainfall cells (> 30 mm/3 h) to each TC center, which is given:

$$DIST30 = \frac{\sum_{i=1}^{N} dist_i \times P_i}{\sum_{i=1}^{N} P_i} \qquad (1)$$

where:

$$P_i = \begin{cases} precip_i, & precip_i \geq 30 \, mm \, per \, 3 \, hours \\ 0, & precip_i < 30 \, mm \, per \, 3 \, hours \end{cases} \qquad (2)$$

where $N$ is grid number within 500 km from the TC center, $precip_i$ is the TC precipitation in the $i$-th grid (unit: mm/3 h), $dist_i$ is the distance between the TC center and the $i$-th grid (unit: km).

The definition of DIST50 is the same as DIST30 but the threshold of heavy rainfall is 50 mm/3 h.

### Trend detection for time series
The linear trends in Figs. 1a, 5, and Supplementary Figs. 1a, 7 are estimated using simple linear regression. Shaded areas in Fig. 1a and Supplementary Fig. 1a are the 95% confidence bounds. In this work, the

significance of linear trends in Fig. 1a and Supplementary Figs. 1a are using the F-test[65] (the **regress()** function and the **LinearModel.fit()** function in Matlab R2016a). The crosses in Fig. 2 indicate the location passed the Mann-Whitney U-test[66] on the regression model of time series based on the 99% significance level (the **ranksum()** function in Matlab R2016a).

## The surface area of grid
Based on the ref. 67, the surface area of each raster is given:

$$A = R^2 \times (\theta_2 - \theta_1) \times (sin\varphi_2 - sin\varphi_1) \tag{3}$$

where $A$ is the surface area (unit: km²), $\theta_1$ and $\theta_2$ is longitude (unit: radians), $\varphi_1$ and $\varphi_2$ is latitude (unit: radians), $R$ is the radius of the earth ($R$ = 6371.39, unit: km). Here we created a mesh of grids boxes with 0.25° × 0.25° by size. Then the monthly mean value of DIST30 is calculated for each grid and used as the response variable for the XGBoost models.

## Land and sea boundary
If a grid cell includes both land and sea, we define the grid cell as the land and sea boundary.

## XGBoost model
Boosting trees is a decision tree-based machine learning approach[33]. The algorithm starts with weak learners, and new trees are generated to reduce the errors of trees in previous rounds. The objective function of XGBoost is written as follows:

$$\mathcal{L}^{(t)} = \sum_{i=1}^{n} l\left(y_i, \hat{y}_i^{(t-1)} + f_t(X_i)\right) + \Omega(f_t) \tag{4}$$

where $\mathcal{L}$ is a differentiable convex loss function measuring the difference between the prediction $\hat{y}_i$ and the target value $y_i$. The term $\Omega$ is a penalization for the complexity of the regression tree functions. The model is trained in an additive manner so $\hat{y}_i^{(t)}$ is the prediction of the $i$-th instance at the $t$-th iteration and $f_t$ is greedily added to minimize the objective $\mathcal{L}$ and improve the model performance. The XGBoost has already been successfully used to reconstruct precipitation isotope records in Europe[68] and historical hurricane wind records[69]. We used the caret R package caret (v 6.0-93) to streamline the model training process of the XGBoost model (R package xgboost v 1.7.3.1). The whole data was sliced into five equal-size samples that are with random data selections. For each data slice, we trained the XGBoost with the remaining 4/5 of data with 5 folds cross-validation. A set of different parameters combinations have been tested (number of rounds, maximum tree depth, subsample ratio of columns when constructing each tree, etc.) using the caret package to achieve the best model fitting, cross-validation, and out-of-sample prediction results. The model is used to predict the observed DIST30 for each 1/5 testing slice. All out-of-sample predictions for five slices are collected and compared with the observation for model predictive performance evaluation. We repeated the process for the entire globe and six ocean basins.

Machine learning models are not only adept at delivering robust prediction performance but can also unveil essential insights into underlying physical processes. One strength of certain machine learning models, like XGBoost, is their ability to identify intricate and non-linear relationships between input features and outputs. To interpret our trained models, we employed the SHapley Additive exPlanations (SHAP) value method, which draws inspiration from Shapley Values in game theory. It offers a systematic mechanism to allocate significance values to individual model features, helping to quantify the influence of each feature on the model's predictions by considering all potential feature combinations. The Shapley value can

be mathematically represented by:

$$\phi_j(v) = \sum_{S \subseteq \{1,\dots,p\}\setminus\{j\}} \frac{|S|!(p-|S|-1)!}{p!} \left(v_x(S \cup \{j\}) - v_x(S)\right) \tag{5}$$

Here, $S$ represents a subset of the $p$ features the model utilizes, and $x$ is the feature value vector of the instance under study. $v_x(S)$ denotes the prediction for feature values in set $S$, marginalized over features outside of set $S$. Calculating the exact SHAP value becomes computationally demanding as it requires evaluating all feature combinations both with and without a particular feature, especially as the feature count grows. To mitigate this computational challenge, approximations like the Monte-Carlo Sampling introduced by Štrumbelj et al. [70] have been proposed. We use the SHAP value to both rank the feature importance and analyze how DIST30 interacts with each individual feature. For our analysis, we harnessed the R package "shapviz" to compute and visualize the SHAP values for all environmental forcing variables in our XGBoost model.

## Data availability
The TCs track dataset is obtained from the International Best Track Archive for Climate Stewardship (IBTrACS) https://www.ncei.noaa.gov/products/international-best-track-archive. Three hourly precipitation data are obtained from the Multi-Source Weighted-Ensemble Precipitation (MSWEP) v2 at http://www.gloh2o.org/. The land and sea boundary shape file are obtained from the UCLA Geoportal at https://guides.library.ucla.edu/gis. Climate/oceanic indexes are obtained from the NOAA Physical Science Laboratory at https://psl.noaa.gov/data/climateindices/list/. The source data generated in this study[71] have been deposited in the https://doi.org/10.5281/zenodo.11190029.

## Code availability
All codes used to read, analyze, and plot the data in this study[71] are available at https://doi.org/10.5281/zenodo.11190029.

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

## Acknowledgements

This study was jointly supported by the National Natural Science Foundation of China (grant no. 72293571 [Chen]), the Faculty Research and Creative Activities Award from Western Michigan University (project no. 2024 [Zhu]), and the National Natural Science Foundation of China (grant no. U22B2011 [Xu]).

## Author contributions

L.Q., L.Z., J.C., and W.X. designed the study. L.Q., L.Z., and Y.T. compiled the data. L.Q., L.Z., and Z.L. conducted the calculations. L.Q., L.Z., B.L., Z.L., S.S., and W.X. analyzed the results. L.Q. and L.Z. created the figures. L.Q. and L.Z. wrote the first draft of the manuscript. L.Q., L.Z., Y.T., G.M., J.C., and W.X. reviewed and edited the manuscript before submission.

## Competing interests

The authors declare no competing interests.
