## [Peer Review File · Nature Communications]

Global Expansion of Tropical Cyclone Precipitation FootprintREVIEWER COMMENTS

Reviewer #1 (Remarks to the Author):

Review recommendation: Major revision

This study investigated the long-term changes in TC precipitation in terms of a spatial scale. In this study, an index called DIST30 was developed to represent the spatial scale of heavy precipitation of TCs. DIST30 showed a significant increase globally, with considerable spatial variability. Factors influencing DIST30 were examined using a XGboost model, and the vertical wind shear was identified as a key factor of long-term changes in DIST30. Recent studies have highlighted long-term changes in TC precipitation based on observational data. Previous studies mainly focused on changes in TC precipitation rates, without detailed characteristics of precipitation structure. Therefore, this study is important in illuminating the detailed characteristics of TC precipitation. However, there are methodological ambiguities hindering the derivation of key conclusions of this study. Considering the importance of the research topic, a major revision is necessary for publication. Detailed comments are provided below.

Main comments:

1. An interpretation of the physical meaning of DIST30 needs to be added to the manuscript. The key findings of this study are based on the DIST30. DIST30 represents the average distance from the TC center weighted by precipitation rates. DIST30 can reflect the complex characteristics of TC precipitation structure. For example, if the same precipitation pattern moves away from the TC center, DIST30 will increase. This means that the intensity and area of precipitation remain constant, but only the position from the TC center changes. On the other hand, if precipitation decreases in the inner-core and increases in the outer regions as described by Tu et al. (2021), DIST30 will increase. It is difficult to understand how the changes in DIST30 mean the changes in precipitation structure of TCs based on the current analysis. Therefore, it is necessary to clearly explain what characteristics of TC precipitation DIST30 can represent and the advantages of using DIST30.
2. In connection with the above comment, it would be beneficial to emphasize the purpose of examining DIST30 further. Previous studies have analyzed the precipitation structure of TCs using various indices. A commonly used index representing the spatial scale of precipitation is areal extent. In addition, the dispersion index (Zick and Matyas 2016; Zhou and Matyas 2021; Kim and Matyas 2024) is very similar to DIST30 but differs in that it is weighted by area rather than precipitation rates. Emphasizing the advantages of DIST30 compared to existing methods would enhance the significance of the research. Zick, S. E. & Matyas, C. J. A Shape metric methodology for studying the evolving geometries of synoptic-scale precipitation patterns in tropical cyclones. *Ann. Am. Assoc. Geogr.* 106, 1217–1235. <https://doi.org/10.1080/24694452.2016.1206460> (2016). Zhou, Y. & Matyas, C. J. Regionalization of precipitation associated with tropical cyclones using spatial metrics and satellite precipitation. *GISci. Remote Sens.* 58, 542–561. (2021). Kim, D., & Matyas, C.J. Classification of tropical cyclone rain patterns using convolutional autoencoder. *Sci Rep* 14, 791 (2024).

3. It would be better to discuss the long-term quality of precipitation data (MSWEP). Is there any difference in the quality between periods that include satellite data such as TRMM/GPM and periods that do not? Is there any potential impact this could have on the analysis of long-term changes?

4. It is necessary to revise the interpretation of trends in DIST30. According to the XGboost model, DIST30 has a negative relationship with VMAX and a positive relationship with VWS. In the tropics, VWS is significantly decreasing, which does not explain the increase in DIST30. Furthermore, SST is increasing in the tropics, which is favorable for the increase in VMAX. That is, if VMAX increases, DIST30 could decrease, which contradicts the findings of this study. Additionally, in the mid-latitudes, while DIST30 is significantly increasing, the increase in VWS is not significant. The increase in the sample number of DIST30 in the mid-latitudes (Fig. 5f) may not fully explain the increase in DIST30, as the annual mean DIST30 in Figure 5d may not be related to sample numbers. Removing linear trends in Figs. 5d and 5f would likely result in a lower correlation between DIST30 and Obs Freq.

Specific comments:

(Abstract) It would be better to provide the definition of DIST30. For example, the mean radial distance of heavy precipitation area weighted by precipitation rate.

(Line 50) There is a typo. "precipitation"

(Line 55) This sentence would be as follows: "TC rainfall rate shows"

(Line 63-66) Several studies also support that the variabilities of TC rain in the inner-core and outer region are different. Here are some relevant references.

Kim, D., Park, D.S.R., Nam, C.C. et al. The parametric hurricane rainfall model with moisture and its application to climate change projections. *npj Clim Atmos Sci* 5, 86 (2022).

Kim, D., Park, D.-S.R. & Matyas, C. J. Spatial variations in tropical cyclone rainfall over the western North Pacific according to ENSO phase. *J. Clim.* 36, 1697–1710. (2023).

(Figs 1b,c) It would be helpful to examine the changes in precipitation rates between the periods of 1980-1999 and 2001-2020, not only focusing on the frequency of heavy precipitation. It will be possible to see how the increase in DIST30 is associated with changes in precipitation structure.

(Fig 2) Only the changes in area of grids where DIST30 increased or decreased was interpreted in the manuscript. However, it is also necessary to interpret the changes in DIST30 magnitude for each ocean basin.

(Line 123) It would be better to define land and sea boundaries more clearly. Do the boundary include land? Or does it mean grids that have fractions of both land and sea?

(Figs 3c-f) The colors in the scatter plot have not been interpreted.

(Line 176-182, 200-201) I agree that VWS can induce asymmetry in TC rain and contribute to the increase

in rain area and DIST30. Additionally, the non-linear relationship between sea surface temperature and DIST30 is consistent with Kim et al. (2021). In Kim et al. (2021), TC rain area increased solely with sea surface temperature in the tropics and was controlled by VWS in the mid-latitudes.

Kim, D., Ho, C.-H., Murakami, H. & Park, D.-S.R. Assessing the influence of large-scale environmental conditions on the rainfall structure of Atlantic tropical cyclones: An observational study. *J. Clim.* 34, 2093–2106. (2021).

Reviewer #2 (Remarks to the Author):

The authors defined a novel metric, DIST30, which represents the footprint of heavy TC precipitation based on a high-resolution satellite precipitation product and the global TC record over the past 41 years. They showed that DIST30 has increased significantly globally at a rate of 0.34 km per year. Spatially, DIST30 increases by 59.87% of the total TC impact area (8.79×10^7 km²), especially in the western North Pacific, the northern Atlantic and the southern Pacific.

The authors also used the XGBoost model, which showed strong performance in both predicting and interpreting DIST30. They also used Shapley Additive exPlanations (SHAP) to interpret the XGBoost model. The results showed that the monthly DIST30 variability is mainly controlled by the variations of the maximum TC wind speed, TC location, sea surface temperature, vertical wind shear and total water column. In particular, the DIST30 shows a very strong positive relationship with vertical wind shear. And more frequent TCs migrating to higher latitudes in the northern hemisphere is the main contributor to the recent global upward trend in DIST30.

The research topic is certainly plausible and important. The manuscript is well-organized and well-written. The novelty of the research is justified. Therefore, I recommend that this manuscript be accepted for publication after a minor revision.

Minor comments:

- v In the methods section, the authors have randomly selected data for the XGBoost model. Would it be wise to use 30 years of data for training and 10 years of data for testing?
- v The authors should provide the architecture of the XGBoost model.
- v RMSE alone is not sufficient to evaluate the performance of the XGBoost model. Authors are encouraged to use more metrics such as mean absolute error, Nash-Sutcliffe efficiency, correlation coefficient, etc.

Reviewer #3 (Remarks to the Author):

Review of "Global Expansion of Tropical Cyclone Precipitation Footprint" by Qin et al.

Summary:

In this manuscript, the authors define a new TC rainfall index, DIST30, to describe the precipitation distribution, and it is found that tropical cyclone (TC) precipitation footprint has shown a significant expansion in recent several decades. The study presents valuable insights into the field of TC precipitation.

Overall, the paper is well organized, and its presentation is good. However, some issues associated with the data, methods and mechanisms still need to be clarified and improved before the manuscript can be considered for publication.

Comments:

1. The use of MSWEP precipitation data is noted, but the predominant reliance on reanalysis data, especially in the earlier periods, raises concerns about the robustness of the findings. It is essential to explore the utilization of alternative satellite-based precipitation datasets (e.g., TRMM or GPM) to validate and supplement the findings.
2. L96: The term "relative frequency of extreme TC rainfall" and the physical significance of DIST30 require clear definition and clarification to enhance reader understanding.
3. L35-36: "Spatially, DIST30 increases by 59.87% of the total TC impact area ($8.79 \times 10^7 \text{ km}^2$)". The statement regarding the increase in DIST30 lacks a clear explanation of how this increase leads to an increase in the total TC impact area. Further clarification on this relationship is necessary for a comprehensive understanding of the results.
4. The "monthly DIST30" calculation and its relationship with various influencing factors need further explanation.
5. The study suggests a strong influence of wind shear on DIST30, but the presented results seem inconsistent with this conclusion. The observed minimal change in DIST30 in tropical regions, despite a significant decrease in vertical wind shear, as well as the conflicting patterns in higher latitudes, require further clarification.
6. L50: "precipitation".
7. L229-230: There is a discrepancy between the figure captions and the figure. The authors should ensure that the captions accurately reflect the content of the respective figure.

REVIEWER COMMENTS

Reviewer #1 (Remarks to the Author):

Review recommendation: Major revision

This study investigated the long-term changes in TC precipitation in terms of a spatial scale. In this study, an index called DIST30 was developed to represent the spatial scale of heavy precipitation of TCs. DIST30 showed a significant increase globally, with considerable spatial variability. Factors influencing DIST30 were examined using a XGboost model, and the vertical wind shear was identified as a key factor of long-term changes in DIST30. Recent studies have highlighted long-term changes in TC precipitation based on observational data. Previous studies mainly focused on changes in TC precipitation rates, without detailed characteristics of precipitation structure. Therefore, this study is important in illuminating the detailed characteristics of TC precipitation. However, there are methodological ambiguities hindering the derivation of key conclusions of this study. Considering the importance of the research topic, a major revision is necessary for publication. Detailed comments are provided below.

[Answer] Many thanks for your comments and suggestions. We have carefully revised our manuscript based on your suggestions. We hope those changes significantly improved the quality of the manuscript.

Main comments:

[Comment #1] An interpretation of the physical meaning of DIST30 needs to be added to the manuscript. The key findings of this study are based on the DIST30. DIST30 represents the average distance from the TC center weighted by precipitation rates. DIST30 can reflect the complex characteristics of TC precipitation structure. For example, if the same precipitation pattern moves away from the TC center, DIST30 will increase. This means that the intensity and area of precipitation remain constant, but only the position from the TC center changes. On the other hand, if precipitation decreases in the inner-core and increases in the outer regions as described by Tu et al. (2021), DIST30 will increase. It is difficult to understand how the changes in DIST30 mean the changes in precipitation structure of TCs based on the current analysis. Therefore, it is necessary to clearly explain what characteristics of TC precipitation DIST30 can represent and the advantages of using DIST30.

[Answer] Thanks for your valuable comment. The physical meaning of DIST30 is the mean radial distance of the area of heavy precipitation of 30 mm per 3 hours weighted by the precipitation rate to the TC center, reflecting the relative position of the extreme TC precipitation to the TC center. The value of DIST30 is determined by precipitation in both the inner core and outer regions, as well as the precipitation structure of the TC. For example, **Comment Fig. 1** presents two precipitation profiles of TC, which are the distance series of rain rate. The profiles have similar precipitation in the inner core and outer regions, but differ in their value of DIST30 due to the different precipitation structure of TC.

DIST30 can partially describe the complexity of TC precipitation structure. Higher value of DIST30 indicates that the most extreme precipitation related to the TC occurs farther away from the center. It is an important indicator to understand potential flood impacts from heavy

TC rainfall. It is important to note that DIST30 is a numerical value calculated solely from the TC precipitation profile (the distance series of TC rain rate) shown in **Comment Fig. 1**. We acknowledge that the full TC precipitation structure still cannot be completely captured by the value of DIST30, which is a limitation of using DIST30. We plan to incorporate the evolution and structure of TC rainfall in our future research.

We have modified the definition of DIST30 in the main text to better reflect its physical meanings in the new text (**lines 33-34, lines 82-85**). We have also included more discussion about the precipitation in both the inner core and outer regions in the discussion section (**lines 343-350**).

Comment Fig. 1. The distance series of TC rain rate. The profiles have similar precipitation in the inner core and outer regions, but differ in their value of DIST30 due to the different precipitation structure of TC.

[Comment #2] Previous studies have analyzed the precipitation structure of TCs using various indices. A commonly used index representing the spatial scale of precipitation is areal extent. In addition, the dispersion index (Zick and Matyas 2016; Zhou and Matyas 2021; Kim and Matyas 2024) is very similar to DIST30 but differs in that it is weighted by area rather than precipitation rates. Emphasizing the advantages of DIST30 compared to existing methods would enhance the significance of the research.

Zick, S. E. & Matyas, C. J. A Shape metric methodology for studying the evolving geometries of synoptic-scale precipitation patterns in tropical cyclones. *Ann. Am. Assoc. Geogr.* 106, 1217–1235. <https://doi.org/10.1080/24694452.2016.1206460> (2016).

Zhou, Y. & Matyas, C. J. Regionalization of precipitation associated with tropical cyclones using spatial metrics and satellite precipitation. *GISci. Remote Sens.* 58, 542–561. (2021).

Kim, D., & Matyas, C.J. Classification of tropical cyclone rain patterns using convolutional autoencoder. *Sci Rep* 14, 791 (2024).

[Answer] Thanks for providing that suggestion. They are great studies and good references for our study. We agree that both the areal extent and the dispersion index are the useful index for representing the spatial scale of precipitation. As mentioned from the comment, DIST30 is similar to the dispersion index in some ways, but differs in that it is weighted by heavy precipitation rate instead of area. We do believe that both the dispersion index (Zick and Matyas, 2016) and our DIST30 provides some perspectives of the spatial distribution of the TC rainfall. In the future, it would be interesting to combine both the area and intensity information in a new index and track their global patterns.

We have added a brief discussion that emphasizes the advances of our DIST30 as compared to existing methods starting from line 85.

[Comment #3] It would be better to discuss the long-term quality of precipitation data (MSWEP). Is there any difference in the quality between periods that include satellite data such as TRMM/GPM and periods that do not? Is there any potential impact this could have on the analysis of long-term changes?

[Answer] Many thanks for raising that important point. In this study, we selected the MSWEP data because it is the longest available high resolution precipitation product with global coverage. It starts from the beginning of satellite age (~1980) and matches with the reliable global TC record. We agree that the MSWEP uses a data fusion approach to merge precipitation data from different sources (satellites, reanalysis, rain gauges, etc) and there are inevitably inconsistencies in those original data sources for MSWEP. Like what you have mentioned, there could be potential impact from those inconsistencies (as the case that TRMM is only available since 1998) in interpreting our long-term changes, but there is no other data source available to validate the trend before the 2000. In order to test the consistency in different data, we finished an independent test based on a different precipitation data (TRMM Multi-satellite Precipitation Analysis Precipitation, TMPA). Results (Comment Fig. 2) show that our findings are consistent based on two different precipitation data after 2000. The DIST30 based on the TMPA data shows an upward trend of 0.20 km/year between 2000 and 2019 (Comment Fig. 2a), with higher relative change in low latitudes (+16.82% beyond 200 km from TC center and -9.24% within 200 km from TC center; Comment Fig. 2b), and in mid latitudes (+0.25% beyond 200 km from TC center and -0.21% within 200 km from TC center; Comment Fig. 2c). In general, those patterns agree with the DIST30 trends/patterns from MSWEP after 2000, as well as the whole upward trend in DIST30 since 1980. We have also added more justification about why we are choosing the MSWEP data in the Data section of Methods, from line 430 to 432.

Comment Fig. 2. As with Fig. 1, but for TMPA precipitation data. a, annually time series of

globally averaged DIST30. b, relative frequency of > 30 mm per 3 hours precipitation in low latitude ($\leq 25^\circ$). c, relative frequency of > 30 mm per 3 hours precipitation in mid-latitude ($> 25^\circ$). Blue lines are for the early period (2000-2009) and red lines are for the late period (2010-2019).

[Comment #4] It is necessary to revise the interpretation of trends in DIST30. According to the XGboost model, DIST30 has a negative relationship with VMAX and a positive relationship with VWS. In the tropics, VWS is significantly decreasing, which does not explain the increase in DIST30. Furthermore, SST is increasing in the tropics, which is favorable for the increase in VMAX. That is, if VMAX increases, DIST30 could decrease, which contradicts the findings of this study. Additionally, in the mid-latitudes, while DIST30 is significantly increasing, the increase in VWS is not significant. The increase in the sample number of DIST30 in the mid-latitudes (Fig. 5f) may not fully explain the increase in DIST30, as the annual mean DIST30 in Figure 5d may not be related to sample numbers. Removing linear trends in Figs. 5d and 5f would likely result in a lower correlation between DIST30 and Obs Freq.

[Answer] Many thanks for the comments and suggestions. In the SHAP analysis of DIST30 (Fig. 3), we can observe a general negative relationship between DIST30 SHAP and VMAX (Fig. 3c), as well as a general negative relationship between DIST30 SHAP and Wind Shear (Fig. 3e). The negative relationship for VMAX is most sensitive when VMAX is below 70 kt, which could be more related to the situation in the midlatitude where the storm intensity decreased but the rain band gets more disbursed. And the curve almost flattened when the VMAX is larger than 70 kt (~category 1 hurricane intensity). So, this indicates that there is a weak DIST30 sensitivity to larger ranges of VMAX, which mostly likely happens over warm ocean water in the tropics. Here we also demonstrate the SHAP value change for SST (included as SI Fig. 5g, here Comment Fig. 3). The SST SHAP also demonstrates different patterns within different ranges, when the SST is below 299K ($\sim 26^\circ\text{C}$, an important threshold for TC genesis and development) there are more positive contributions dominate (corresponding to larger DIST30), but there are both positive and negative contributions to DIST30 when the SST is above 299K. Both relationships for VMAX and SST indicate a complex non-linear relationship between VMAX and DIST30, particularly for well-developed TCs over the tropics. The relationship for Wind Shear's SHAP is also highly non-linear, and generally agrees with the relationships in VMAX and SST. That is, when the wind shear is lower (< 0 kt), the SST is higher (red color in Fig. 3d), there are both positive and negative contributions from the wind shear to DIST30, but the averaged contribution is negative (black line below zero) and smaller DIST30 is corresponding to this part of data. But when the wind shear is much larger, RSST drops to much lower ranges in higher latitude, and most of the SHAP contributions from wind shear are positive for this range.

In Fig. 5a, the trend of DIST30 in the midlatitude could be influenced by both observation frequency and wind shear. The increasing trends in DIST30 and Obs Freq for mid latitudes generally agree with each other after 1990, except that the Obs Freq has a sudden dip in year 2009. The 2009 dip significantly changed the 5 year moving average curve but does not change the general increasing trend in the Obs Freq. But the wind shear time series does not have this sudden drop in 2009, instead the wind shear has a spell of above average observations after 2009. As indicated from the SHAP plot, the DIST30 has high sensitivity to vertical shear in mid

latitudes, The increasing trend in DIST30 has not been influenced significantly by the one big drop of Obs Freq in 2009, the slightly increased wind shear could play an important role particularly after 2009. We also created a linear regression (**Comment Fig. 4**) for the 5-year smoothed DIST30 in the northern hemisphere based on the 5-year smoothed WS and Obs Freq. It indicates that the WS and Freq can capture the DIST30 variability well from a simple linear model. And estimated parameters for both WS and Obs Freq are both positive. **Comment Fig. 4** also indicates the mid-latitude DIST30 in northern hemisphere is generally positive correlated with both the vertical wind shear and the frequency of observation. We have made some modifications in the original description and discussions in the manuscript (**multiple places in result and conclusion sections**).

Comment Fig. 3. The SHAP value for SST.

Comment Fig. 4. The fitted vs observed the smoothed DIST30 in the mid-latitudes of northern hemisphere, linear model is constructed based on smoothed Wind Shear (WS) and the observed frequency (OBSFreq). The linear model function for the DIST30 is displayed in the figure, with R^2 displayed between the observed DIST30 and fitted DIST30. Black line is a linear line fitted the scatter dots (black) and red dash line is for $y=x$ line.

Specific comments:

[Comment #5] (Abstract) It would be better to provide the definition of DIST30. For example, the mean radial distance of heavy precipitation area weighted by precipitation rate.

[Answer] Thanks for your valuable suggestion. We have updated the definition of DIST30 in Abstract as well as its physical meaning **from line 33 to line 37**: “Here we define a novel metric, DIST30, as the mean radial distance from centers of clustered heavy rainfall cells (> 30 mm/3 hours) to the TC center. DIST30 represents the footprint of heavy TC precipitation based on high-resolution precipitation data and the global TC record over the past 41 years for which satellite observations are available.”

[Comment #6] (Line 50) There is a typo. “precipitation”

[Answer] Thanks for your careful reading. We have corrected this typo.

[Comment #7] (Line 55) This sentence would be as follows: “TC rainfall rate shows”

[Answer] Thanks for your suggestion. We have updated this sentence “However, recent studies based on satellite rainfall measurements^{12,13} demonstrate that the TC rainfall rate shows a decreasing trend in the inner-core of the rainband with increases in the outer bands.”

[Comment #8] (Line 63-66) Several studies also support that the variabilities of TC rain in the inner-core and outer region are different. Here are some relevant references.

Kim, D., Park, D.S.R., Nam, C.C. et al. The parametric hurricane rainfall model with moisture and its application to climate change projections. *npj Clim Atmos Sci* 5, 86 (2022).

Kim, D., Park, D.-S.R. & Matyas, C. J. Spatial variations in tropical cyclone rainfall over the western North Pacific according to ENSO phase. *J. Clim.* 36, 1697–1710. (2023).

[Answer] Thanks for your recommendation. They are great studies and good references for our study. We have added the new sentence “Recent studies^{19,20} also find that TC precipitation in the inner core and outer region is controlled by TC intensity and environmental conditions” in the Introduction section.

[Comment #9] (Figs 1b, c) It would be helpful to examine the changes in precipitation rates between the periods of 1980-1999 and 2001-2020, not only focusing on the frequency of heavy precipitation. It will be possible to see how the increase in DIST30 is associated with changes in precipitation structure.

[Answer] Thanks for your valuable comment. Following your suggestion, we have examined

the changes in rain rates between the periods 1980-1999 and 2001-2020 (**Comment Fig. 5**). It was observed that rain rates decreased more during the latter period (2001-2020) at the location closer to the center of the TC compared to the previous period (1980-1999), which is roughly consistent with the findings of Tu et al. (2021). The changes in precipitation structure between the two time periods can partially explain the rise in DIST30.

In addition, according to the physical meaning of DIST30, we also plot the distance series of total heavy precipitation (> 30 mm per 3 hours) in **Comment Fig. 6**. We observe that the changes in heavy precipitation structure between the two time periods are similar to the relative frequency of precipitation exceeding 30 mm per 3 hours in **Fig. 1b-c**. This observation can also explain the increase in DIST30.

Comment Fig. 5. The distance series of mean TC rain rates between the periods of 1980-1999 and 2001-2020. a, in low latitude ($\leq 25^\circ$). b, in high latitude ($> 25^\circ$). Blue lines are for the early period (1980-1999) and red lines are for the late period (2001-2020).

Comment Fig. 6. As with **Comment Fig. 5**, but the sum of rain that exceeds 30 mm per 3 hours. a, in low latitude ($\leq 25^\circ$). b, in mid latitude ($> 25^\circ$). Blue lines are for the early period (1980-1999) and red lines are for the late period (2001-2020).

[Comment #10] (Fig 2) Only the changes in area of grids where DIST30 increased or decreased was interpreted in the manuscript. However, it is also necessary to interpret the changes in DIST30 magnitude for each ocean basin.

[Answer] Thanks for your valuable suggestion. We have listed the details of the areas of magnitude of the differences in DIST30 between the late period (2001-2020) and the early period (1980-1999) in **Supplementary Table 2** by following your suggestion. Some descriptions are added **between line 134 and 137**.

[Comment #11] (Line 123) It would be better to define land and sea boundaries more clearly. Do the boundary include land? Or does it mean grids that have fractions of both land and sea?

[Answer] Thanks for your valuable point. Yes, if a grid has the fractions of both land and sea, we define the grid as the land and sea boundary in this study. A more detailed description has been added in **line 151**.

[Comment #12] (Figs 3c-f) The colors in the scatter plot have not been interpreted.

[Answer] Thanks for raising this important point. The color in the SHAP scatter plot means the second independent variable that has the largest covariance with the main independent variable as in the x axis. Some variable colors have complex relationships with the main independent variable and hard to interpret physically, such as the Atlantic Multidecadal Oscillation (AMO) and VMAX in **Fig. 3c**. Some variable colors actually provide valuable additional information for the physical interpretation of the main independent variable, such as the relative sea surface temperature (RSST) and VS in **Fig. 3e**. We have added more information in the figure caption and some revised discussion in the text.

[Comment #13] (Line 176-182, 200-201) I agree that VWS can induce asymmetry in TC rain and contribute to the increase in rain area and DIST30. Additionally, the non-linear relationship between sea surface temperature and DIST30 is consistent with Kim et al. (2021). In Kim et al. (2021), TC rain area increased solely with sea surface temperature in the tropics and was controlled by VWS in the mid-latitudes.

Kim, D., Ho, C.-H., Murakami, H. & Park, D.-S.R. Assessing the influence of large-scale environmental conditions on the rainfall structure of Atlantic tropical cyclones: An observational study. *J. Clim.* 34, 2093–2106. (2021).

[Answer] Thanks for your valuable comments, we believe the relationships between VWS and DIST30 agrees with the previous findings for the TC rain area, particularly in mid-latitudes. We also added the suggested literature in our discussion.

Reviewer #2 (Remarks to the Author):

The authors defined a novel metric, DIST30, which represents the footprint of heavy TC precipitation based on a high-resolution satellite precipitation product and the global TC record over the past 41 years. They showed that DIST30 has increased significantly globally at a rate of 0.34 km per year. Spatially, DIST30 increases by 59.87% of the total TC impact area (8.79×10^7 km²), especially in the western North Pacific, the northern Atlantic and the southern Pacific.

The authors also used the XGBoost model, which showed strong performance in both predicting and interpreting DIST30. They also used Shapley Additive exPlanations (SHAP) to interpret the XGBoost model. The results showed that the monthly DIST30 variability is mainly controlled by the variations of the maximum TC wind speed, TC location, sea surface temperature, vertical wind shear and total water column. In particular, the DIST30 shows a very strong positive relationship with vertical wind shear. And more frequent TCs migrating to higher latitudes in the northern hemisphere is the main contributor to the recent global upward trend in DIST30.

The research topic is certainly plausible and important. The manuscript is well-organized and well-written. The novelty of the research is justified. Therefore, I recommend that this manuscript be accepted for publication after a minor revision.

[Answer] Many thanks for your comments and suggestions. We have addressed all of them point by point and hope they improve the clarity and quality of the manuscript.

Minor comments:

[Comment #1] In the methods section, the authors have randomly selected data for the XGBoost model. Would it be wise to use 30 years of data for training and 10 years of data for testing?

[Answer] Thanks for the comment and the valuable question. We have identified an increasing trend in DIST30 in 41 years of data and indicate there could be temporal changes in physical processes that control the DIST30 variability. So, the DIST30 data is non-stationary, and we will not be able to get very good out of sample prediction performance if we are using previous 31 years of data to train the model and last 10 years for testing. There will also potential under-sample issues because there is large uncertainty in annual climate variabilities and TC locations. Our main purpose for this article is to develop XGBoost model that can identify important factors/processes that possibly control the monthly DIST30 variability and have decent amount of out-of-sample prediction skills to avoid over-fitting. However, the current model may not capture the non-stationarity in the time series and the complex inter-annual changes in global TC activities very well. We have trained two new XGBoost models, one based on 1980-2010 data and the other based on 1990-2020 data. Then we tested the model prediction performance using 2011-2020 data and 1980-1989 data, respectively. Although both models have good performance when predicting data within their time range (with $R^2 > 0.5$), **Comment Fig. 7 a&b** showed that their prediction performance for years never appeared in the model deteriorate from the randomly selected 5 slice hold-out cross validation (**Fig. 3a**) because of the non-

stationarity in processes controlling the DIST30. Note that the LAT and WS demonstrates higher importance rank in the 1990-2020 model than 1980-2010 model, this is consistent with our argument that increases in TCs moving to the mid-latitudes and wind shear change in the mid-latitudes in the northern hemisphere after 1990 play important role in the increasing DIST30 trend globally. Therefore, challenges are still existed in training a ML model to skillfully predict possible future changes in DIST30 under climate change, as well as their annual variabilities. Since DIST30 is a feature of TC, the future works need to be first focused on developing ML models that can predict annual variabilities of regional and global TC genesis and locations well.

Comment Fig. 7. a. prediction of DIST30 for 2011-2020 by using a XGBoost model trained by data of 1980-2010; b. prediction of DIST30 for 1980-1989 by using a XGBoost model trained by data of 1990-2020. c. feature importance calculated as the SHAP value for eight top ranked features with the most importance for the XGBoost model trained by data of 1980-2010; d. feature importance calculated as the SHAP value for eight top ranked features with the most importance for the XGBoost model trained by data of 1990-2020.

[Comment #2] The authors should provide the architecture of the XGBoost model.

[Answer] Many thanks for your suggestions, we have added more description about the XGBoost model and the SHAP value in the new manuscript to improve the clarity.

[Comment #3] RMSE alone is not sufficient to evaluate the performance of the XGBoost model. Authors are encouraged to use more metrics such as mean absolute error, Nash-Sutcliffe efficiency, correlation coefficient, etc.

[Answer] Thanks for your valuable suggestion, we have added the MAE in the new **Fig. 3** to provide more robust metrics for model evaluation. We have included the R^2 already, which is a squared correlation coefficient. Since the focus here is not to compare different ML approaches, we will save Nash-Sutcliffe efficiency and other metrics for future ML algorithm comparison studies.

Reviewer #3 (Remarks to the Author):

Review of “Global Expansion of Tropical Cyclone Precipitation Footprint” by Qin et al.

Summary:

In this manuscript, the authors define a new TC rainfall index, DIST30, to describe the precipitation distribution, and it is found that tropical cyclone (TC) precipitation footprint has shown a significant expansion in recent several decades. The study presents valuable insights into the field of TC precipitation.

Overall, the paper is well organized, and its presentation is good. However, some issues associated with the data, methods and mechanisms still need to be clarified and improved before the manuscript can be considered for publication.

[Answer] Many thanks for your comments. We have carefully edited the corresponding parts and hopefully those further improved the quality and clarity of the presentation.

Comments:

[Comment #1] The use of MSWEP precipitation data is noted, but the predominant reliance on reanalysis data, especially in the earlier periods, raises concerns about the robustness of the findings. It is essential to explore the utilization of alternative satellite-based precipitation datasets (e.g., TRMM or GPM) to validate and supplement the findings.

[Answer] Thanks for your valuable suggestion. In the latest version, we have discussed the results based on a different precipitation data (TRMM Multi-satellite Precipitation Analysis (TMPA) Precipitation) by following your suggestion. Our new results (**Comment Fig. 8**) show that our findings are consistent based on two different precipitation data between 2000 and 2019. The DIST30 with the TMPA data shows an upward trend of 0.20 km/year (**Comment Fig. 8a**) in the 20 years that are available, which is similar to the 0.34 km/year trend based on 41 years of MSWEP data. There are higher changes for relative rain rate frequency in low latitudes (+16.82% beyond 200 km from TC center and -9.24% within 200 km from TC center, **Comment Fig. 8b**), but lower changes in mid latitudes (+0.25% beyond 200 km from TC center and -0.21% within 200 km from TC center; **Comment Fig. 8c**). We hope this extra analysis solidifies the findings that we had previously. In addition, we have also added more justification about why we are choosing the MSWEP data in the Data section of Methods, **from line 430 to 432**.

Comment Fig. 8. As with Fig. 1, but for TMPA precipitation data. a, annually time series of globally averaged DIST30. b, relative frequency of > 30 mm per 3 hours precipitation in low latitude ($\leq 25^\circ$). c, relative frequency of > 30 mm per 3 hours precipitation in mid-latitude ($> 25^\circ$). Blue lines are for the early period (2000-2009) and red lines are for the late period (2010-2019).

[Comment #2] L96: The term "relative frequency of extreme TC rainfall" and the physical significance of DIST30 require clear definition and clarification to enhance reader understanding.

[Answer] Thanks for your valuable comment. In Fig. 1b&c, the x-axis indicates the distance from the center of the TC, while the y-axis represents the relative frequency of extreme precipitation at that distance occurring over a given time period. If the relative frequency of extreme precipitation occurrences over small distances decreases, and the relative frequency of extreme precipitation occurrences over large distances increases, the DIST30 calculated by Equation 1 becomes larger.

[Comment #3] L35-36: "Spatially, DIST30 increases by 59.87% of the total TC impact area ($8.79 \times 10^7 \text{ km}^2$)". The statement regarding the increase in DIST30 lacks a clear explanation of how this increase leads to an increase in the total TC impact area. Further clarification on this relationship is necessary for a comprehensive understanding of the results.

[Answer] Thanks for raising that point. Here the majority areas experienced increases in DIST30, indicating more areas are experiencing elevated risk of heavy precipitation far away from the TC center. We added some clarification for the meaning of our findings in line 158 to 161.

[Comment #4] The "monthly DIST30" calculation and its relationship with various influencing factors need further explanation.

[Answer] Thanks for your question. In this study, the monthly DIST30 is calculated as the average of all DIST30s for each month within each $0.25^\circ \times 0.25^\circ$ boxes, we added descriptions in the captions of Fig. 3 and method section in line 467.

[Comment #5] The study suggests a strong influence of wind shear on DIST30, but the presented results seem inconsistent with this conclusion. The observed minimal change in DIST30 in tropical regions, despite a significant decrease in vertical wind shear, as well as the

conflicting patterns in higher latitudes, require further clarification.

[Answer] Thanks for raising that point. Yes, we agree that relationships between the wind shear and DIST30 are complex and nonlinear (**Fig. 3e**). In the tropics, the wind shear (WS) is generally lower and the DIST30 has low sensitivity to vertical shear in the tropics, with both positive and negative contributions to DIST30 (**Fig. 3e**, when WS is below 15 knots). So, the DIST30 is less sensitive to changes in WS in the tropics and reduced wind shear in the tropics does not significantly decrease the DIST30 over the tropics. However, generally theories mentioned that reduced WS in the tropics will likely benefit TC genesis and growth. In the mid-latitudes, the DIST30 has higher sensitivity to the wind shear following a general positive relationship. As more TCs entering midlatitude and a slightly increased WS in midlatitude, we are observing a strong increasing trend of DIST30 in midlatitudes in the northern Hemisphere.

[Comment #6] L50: "precipitation".

[Answer] Thanks for your careful reading. We have corrected this typo.

[Comment #7] L229-230: There is a discrepancy between the figure captions and the figure. The authors should ensure that the captions accurately reflect the content of the respective figure.

[Answer] Thanks for your careful reading. We apologize for the discrepancy between the captions and the figure in **Fig. 4**. In the latest version, we have updated the captions in **Fig. 4c&d**.

REVIEWERS' COMMENTS

Reviewer #1 (Remarks to the Author):

Review recommendation: Accept

The authors have appropriately answered the questions raised and accordingly revised the manuscript. Therefore, I recommend the publication of this paper. Please correct the typos found throughout.

Reviewer #2 (Remarks to the Author):

I recommend this manuscript for publication.

Reviewer #3 (Remarks to the Author):

Second review of "Global Expansion of Tropical Cyclone Precipitation Footprint" by Qin et al. (NCOMMS-24-06089A)

I greatly appreciate the authors' substantial revisions and detailed responses to my earlier concerns, and I am satisfied with the current revised version. I recommend that this manuscript be accepted for publication.